# Multiple Fano Resonances with Tunable Electromagnetic Properties in Graphene Plasmonic Metamolecules

**DOI:** 10.3390/nano10020236

**Published:** 2020-01-29

**Authors:** Hengjie Zhou, Shaojian Su, Weibin Qiu, Zeyang Zhao, Zhili Lin, Pingping Qiu, Qiang Kan

**Affiliations:** 1Fujian Key Laboratory of Light Propagation and Transformation, College of Information, Science and Engineering, Huaqiao University, Xiamen 361021, China; hjzhou@stu.hqu.edu.cn (H.Z.); sushaojian@hqu.edu.cn (S.S.); zyzhao@stu.hqu.edu.cn (Z.Z.); zhllin@hqu.edu.cn (Z.L.); 2Fujian Key Laboratory of Semiconductor Materials and Applications, Xiamen University, Xiamen 361005, China; 3College of Materials Science and Opto-Electronic Technology, University of Chinese Academy of Sciences, Beijing 100086, China; ppqiu@semi.ac.cn (P.Q.); kanqiang@semi.ac.cn (Q.K.); 4Institute of Semiconductors, Chinese Academy of Sciences, Beijing 100086, China

**Keywords:** Fano resonances, graphene metamolecule, chemical potential, surface plasmon, plasmonic coupling, modulation depth

## Abstract

Multiple Fano resonances (FRs) can be produced by destroying the symmetry of structure or adding additional nanoparticles without changing the spatial symmetry, which has been proved in noble metal structures. However, due to the disadvantages of low modulation depth, large damping rate, and broadband spectral responses, many resonance applications are limited. In this research paper, we propose a graphene plasmonic metamolecule (PMM) by adding an additional 12 nanodiscs around a graphene heptamer, where two Fano resonance modes with different wavelengths are observed in the extinction spectrum. The competition between the two FRs as well as the modulation depth of each FR is investigated by varying the materials and the geometrical parameters of the nanostructure. A simple trimer model, which emulates the radical distribution of the PMM, is employed to understand the electromagnetic field behaviors during the variation of the parameters. Our proposed graphene nanostructures might find significant applications in the fields of single molecule detection, chemical or biochemical sensing, and nanoantenna.

## 1. Introduction

Noble metal nanoparticles, such as Au and Ag, interact strongly with the incident light and cause the collective oscillations of the electrons, which further result in localized surface plasmon resonances (LSPRs) [1,2]. LSPR is sensitive to the size of the nanoparticle (NP). At the same time, variation in the geometrical structures and the properties of the surrounding medium also have a great impact on the excitation effect of LSPR [3]. In recent years, NPs with different shapes and sizes have been widely researched in LSPR, such as nanorings [2,4], nanodiscs [5,6], nanospheres [7], mismatched NP dimer [8], nanorod [9], nanoshells [10,11], and NanoCross [12]. Simultaneously, in plasmonic molecules (PMs) composed of NPs, the unique hybridized plasmon modes coupled by LSPRs raise a wide range of optical phenomena like electromagnetically induced transparency (EIT) [13,14,15,16] and Fano resonances (FRs) [13,17,18,19]. FR with high local field enhancement [4] results from the coherent interference of the super-radiant bright mode and the subradiant dark mode [5,20], which has gained attention in noble metal PMs [21,22,23]. At the same time, the independent subgroup mode is also used to understand the line shape of FR [6,17,24]. Usually, several modes are generated by breaking the spatial symmetry of the metal nanostructure, which makes it possible to observe multiple FRs in the spectrum [13,25,26]. Compared with a single FR, the line shape of multiple FRs can be simultaneously modified by adjusting geometric parameters. Therefor multiple FRs are applied in multi-wavelength surface-enhanced Raman spectroscopy (SERS) [2,27]. However, due to the high ohmic loss and the obstacle in varying permittivity of noble metals [28,29], the quality of LSPR is degraded [30]. Furthermore, the working wavelength is unchangeable once the geometrical structures of the PMs are ascertained, which restricts the feasible applications in the fields of light-matter interaction, nanoantenna, chemical sensing, and biochemical sensing [31,32,33].

Graphene-supported plasmons offer excellent properties such as low damping and intrinsically ultra-high confinement due to the unique Dirac conical band structure, which overcome the limits of noble metals [34,35,36,37]. Therefore, graphene plasmons (GPs) have been widely employed in polarization control devices [38,39], optical modulators [40,41], and sensors [42,43]. Furthermore, the working frequency range from near-infrared to terahertz of GPs is electrically and chemically tunable, which enables devices based on GPs to be more widely applied [37,44,45]. GPs in the infrared region have been applied in the fields of surface-enhanced infrared absorption (SEIRA) and light trapping [42,46]. At the same time, the FR produced by GPs have a sharp spectra response. Moreover, due to the adjustable chemical potential of graphene, the Fano resonance produced by graphene has a wider application [28].

In this research paper, the graphene plasmonic metamolecule with a multiple ring structure is proposed to generate multiple FRs. We find that there are two FRs in the extinction spectrum by adding an additional 12-ring nanodisc to a heptamer without changing the symmetrical nature. Two FR modes with different wavelengths are obtained in the extinction spectrum of the proposed plasmonic metamolecule. The competition and switching nature of these two FR modes are investigated by varying the material and geometrical parameters of the central, intermediate ring and outer ring nanodiscs. Furthermore, a simple trimer model is employed to emulate the radical direction of the whole structure, and the mechanism of the properties of the FR modes is explored. The proposed graphene nano-structure with the excellent properties may find broad area applications in the fields of light-matter interaction, chemical and biochemical sensing, nanoantenna, Surface-Enhanced Raman Scattering (SERS), and single molecule detection.

## 2. Calculation Methods and Models

The proposed PM consisting of 19 graphene nanodiscs with a *D_6h_* point group symmetry is placed on a calcium fluoride (CaF_2_) substrate, which is schematically demonstrated in Figure 1a. In the infrared region, SiO_2_ substrate produces strong phonon-related resonance peaks, which strongly hybridize with graphene plasmon modes [42]. Therefore, these phonons make graphene plasmonic field enhancement decrease drastically [47,48]. Because of the broadband transparency of CaF_2_ in the mid-infrared (MIR) region, the substrate phonon effect is eliminated by using CaF_2_ as a substrate [4,42]. In addition, the 19 graphene nanodiscs keep the *D_6h_* point group symmetry, as shown in Figure 1b. It is worth noting that the metamolecule are divided into three parts: the central, the intermediate ring, and the outer ring nanodiscs. For convenience, the radii of the central nanodisc, the intermediate ring, and the outer ring nanodiscs are labelled as $R_{1}$, $R_{2},$ and $R_{3}.$ The corresponding chemical potentials are marked as $\mu_{c1}$, $\mu_{c2}$, and $\mu_{c3}$, respectively. COMSOL Multi-Physics, which is a commercial finite element method (FEM) software, is used to calculate the electromagnetic fields and the extinction spectra of graphene metamolecules in this study.

Graphene, with only one layer of monoatomic thickness, is modelled as a material with equivalent relative permittivity. The equivalent relative permittivity is calculated by the equation below.
(1)$$\varepsilon =1+\frac{i\sigma_{g}\eta_{0}}{k_{0}\Delta }$$
where $k_{0}=2\pi /\lambda$ is the wavenumber of light in free space, the graphene layer thickness is Δ = 0.334 nm, and $\eta_{0}$ = 377 Ω, which stands for the impedance of free space. In Equation (1), the equivalent relative permittivity (ε) is decided by the surface conductivity $\sigma_{g}$. The surface conductivity is given by the equation below.
(2)$$\begin{array}{l} \sigma_{g}=\frac{2e^{2}k_{B}T}{\pi \hbar^{2}}\cdot \frac{i}{\omega +i\tau^{-1}}\left[\ln\left(2\cosh\left(\frac{\mu_{c}}{k_{B}T}\right)\right)\right]\\ +\frac{e^{2}}{4\hbar }\left[\frac{\sinh(\frac{\hbar \omega }{2k_{B}T})}{\cosh\left(\frac{\mu_{c}}{k_{B}T}\right)+\cosh\left(\frac{\hbar \omega }{2k_{B}T}\right)}-\frac{i}{2\pi }\ln\frac{{\left(\hbar \omega +2\mu_{c}\right)}^{2}}{{\left(\hbar \omega -2\mu_{c}\right)}^{2}+{\left(2k_{B}T\right)}^{2}}\right] \end{array}$$
where *e* is the charge of an electron, $k_{B}$ is the Boltzmann’s constant, ℏ is the reduced Planck constant, $\mu_{c}$ is the chemical potential (i.e., Fermi energy) of the doped graphene, which is obtained by $\mu_{c}=\hbar v_{f}\sqrt{\pi n}$, where *n* is the carrier density and $v_{f}=c/300$ represents the Fermi velocity, ω=2πυ is the angular frequency, and $\tau =\mu \mu_{c}/ev_{f}^{2}$ is the momentum relaxation time, where $\mu =10000\;cm^{2}V^{-1}s^{-1}$ indicates the carrier mobility of graphene. At room temperature, the absolute temperature (*T*) is set as 300K. In this equation, the first-term stands for intra-band electron-photon scattering, and the second term represents an inter-band electron-electron transition. The extinction spectra are expressed by the extinction cross-section $\sigma_{\mathrm{ext}}$.
(3)$$\sigma_{ext}=\sigma_{abs}+\sigma_{sc}$$
where the absorption cross-section $\sigma_{abs}$ is described as
(4)$$\sigma_{abs}=\frac{1}{I_{0}}∭QdV$$
and the scattering cross-section $\sigma_{sc}$ is given by
(5)$$\sigma_{sc}=\frac{1}{I_{0}}\iint \left(\vec{n}\cdot \vec{S_{sc}}\right)dS$$

In these equations, $I_{0}$ is the incident intensity. $\vec{n}$ represents the normal vector that points outwards from the graphene plasmonic nanocluster. $\vec{S_{sc}}$ corresponds to the scattered intensity electromagnetic energy intensity. Q stands for the power loss density in the graphene metamolecule. In Equation (4), the integral is taken over its volume. In Equation (5), the integral is taken over the closed surface of the scatter. The thickness of the graphene nanodisc is meshed at least 10 layers to ensure the accuracy of the result. Simultaneously, the Perfectly Matched Layer (PML) is set around the nanostructure to prevent the reflected light fields.

## 3. Results and Discussions

### 3.1. Fano Resonance Modes Induced by the Central Nanodisc and the Ring Nanodiscs

FRs are obtained by two approaches. One approach is to destroy the symmetry of structures, such as split nanorings [2], dual-disk ring plasmonic nanostructures [26], removing a wedge from a nanodisc [5], and split-ring resonator/disk nanocavities [49]. Another approach is to introduce additional nanoparticles without degrading the spatial symmetry, such as the double rings nanostructure [50], central nanoparticle and ring nanoparticles [6,7], and adding nanorods in the ring [9]. In this paper, we investigate the mechanism of FR modes by adding additional nanodiscs without reducing the spatial point group symmetry. In this case, a nanostructure, which consists of 19 nanodiscs (a central nanodisc and two ring nanodiscs), is considered to study the interaction of nanodiscs in FRs. The radius of the central nanodisc is $R_{1}$ = 90 nm, and other nanodiscs are set as 50 nm. The chemical potentials of the central nanodisc and the ring nanodiscs are 0.6 eV and 0.5 eV, respectively. In order to evaluate the influences of the ring nanodiscs on FRs, the structure is divided into two parts, i.e., the heptamer and the outer ring nanodiscs. In the first part, the central nanodisc is treated as an additional nanodisc. Extinction spectra of the hexamer (number of nanodiscs N = 6) and the heptamer (N = 7) are shown in Figure 2a. It can be seen that only one peak exists in the extinction spectrum of the hexamer. However, the line shape of the FR mode with two peaks and a dip appears when a central nanodisc is added into the hexamer, which can be demonstrated by two subgroup modes (the ring mode and the central mode) in our previous study [24]. Based on this nanostructure, additional ring nanodiscs (N = 12) with the chemical potential of 0.5 eV are added around the heptamer to constitute the PMMs (N = 19) for the ultimate research target. In this case, the extinction spectra of the outer ring nanodiscs (N = 12) and the whole nanostructure (N = 19) are presented in Figure 2b. It is clear that an independent peak exists in the extinction spectrum when N = 12, which is similar to the case of the hexamer (N = 6) composed of six nanodiscs. Intriguingly, three peaks and two dips appear when this heptamer is added into ring nanodiscs (N = 12), which is similar to the case of hexamer and heptamer, but another additional peak and dip appear. In previous studies, multiple FRs are obtained by adding additional ring NPs without varying the symmetry [9,51]. To explain the appearance of multiple FRs, the near-field distributions of the plasmonic peaks for the heptamer (N = 7), and three plasmonic peaks for the whole structure (N = 19) are presented in Figure 2c. In the extinction spectrum of the heptamer, two peaks are labelled as a and b, respectively. On the other hand, similar to the case of heptamer, three peaks in the extinction spectrum of the whole metamolecule are marked as A, B, and C, respectively. For peak a, it can be clearly seen that hot spots mainly distribute on the edge of all nanodiscs and the main contributions are the two top nanodiscs and the two bottom nanodiscs, as shown in Figure 2c. However, for peak b, hot spots move from the edge to the gap between the ring nanodiscs and the central nanodisc, which mainly attributes to the six-ring nanodiscs. Intriguingly, in the near-field distribution of peak C, the main contributions from the middle seven nanodiscs are similar to peak b, while the main contributions of peak a are alike to the middle seven nanodiscs of peak B. Therefore, the lower energy FR mode is original, while the higher energy FR mode is generated by adding the outer ring nanodiscs. For convenience, the lower energy (longer wavelength) FR mode is labeled as FR_1_ mode, and the higher energy (shorter wavelength) FR mode is marked as FR_2_ mode. For the near-field distribution of peak C, the main contributions are generated by the intermediate ring nanodiscs. Furthermore, it is apparent that the outer ring nanodiscs couple weakly with the heptamer. However, peak A and peak B are opposite to peak C. In peak B, the main contributions come from the outer ring nanodiscs. However, compared with the heptamer, the electromagnetic field of the middle seven nanodiscs does not vary. For peak A, hot spots of the top two nanodiscs and the bottom two nanodiscs at the intermediate ring and the top middle nanodisc and the bottom middle nanodisc at the outer ring are stronger than other nanodiscs. Meanwhile, this indicates that strong coupling between the intermediate ring nanodiscs and the outer ring nanodiscs occurs in peak A. Strikingly, it is found that the coupling between the outer ring nanodiscs and the middle seven nanodiscs is enhanced gradually from peak C to peak A. Thus, the FR_1_ mode is generated in the case of weak coupling between the outer ring nanodiscs and the middle seven nanodiscs, while the case of FR_2_ mode is opposed to the FR_1_ mode. Therefore, the near-field distribution of peak B consists of the original mode of the heptamer and the mode of coupling between the outer ring nanodiscs and the heptamer, which results in the amplitude of peak B being higher than peak C.

### 3.2. The Relationship between FRs and the Outer Ring Nanodiscs

Next, our attention is switched to the relationship between the outer ring nanodiscs and the FRs, which is worth studying. In order to preserve spatial symmetry of the structure, we simultaneously vary the chemical potential or the radius of 12 nanodiscs in the outer ring. The resonant frequency of PMM is regulated by the chemical potential of graphene, which is impossible in noble metals if the nanodiscs are fixed. We adjust the chemical potential ($\mu_{c3}$) of the outer ring nanodiscs from 0.5 eV to 0.6 eV, whereas the chemical potential of graphene is tunable by two approaches, i.e., electrostatic and chemical doping. The chemical potential of graphene is related to the carrier concentration [52].
(6)$$n_{s}=\frac{2}{\pi \hbar^{2}v_{f}^{2}}\int_{0}^{\infty }\varepsilon \left[f_{d}\left(\varepsilon -\mu_{c}\right)-f_{d}\left(\varepsilon +\mu_{c}\right)\right]d\varepsilon$$
where $f_{d}=1/\left\{1+exp\left[\left(\varepsilon -\mu_{c}\right)/\left(K_{B}T\right)\right]\right\}$ is Fermi-Dirac distribution and ε is the energy. These two approaches are realized by adjusting the carrier concentration of graphene and further controlling the chemical potential of the graphene nanodisc. For electrostatic doping, a top gate configuration textures the chemical potential of the graphene nanodisc by supplying appropriate top gate voltage locally [30,52,53]. For chemical doping, the graphene nanodiscs are exposed to the nitric acid vapour or nitrogen dioxide while other nanodiscs should be covered by a layer of the diaphragm [48,54]. The extinction spectra with the change of $\mu_{c3}$ are plotted in Figure 3a,b. In Figure 3a, it is observed that the amplitude of peak A decreases gradually as $\mu_{c3}$ varies from 0.5 eV to 0.52eV. Strikingly, the FR_2_ mode disappears completely when $\mu_{c3}$ is 0.52 eV, while the FR_1_ mode still exists and has a blue shift in the spectra. Simultaneously, the amplitude of peak B is lower than peak C. As $\mu_{c3}$ continues to increase, the FR_1_ mode also begins to disappear.

A simple trimer model is constructed to emulate the relative distribution of the nanostructure in the radiative direction, as shown in Figure 3c. This helps understand the effect on the other two sets of nanodiscs with the change of chemical potential of the outer ring nanodiscs. In the PMM, it can be clearly seen that the distribution of chemical potential is symmetrical about the central nanodisc, and the chemical potentials are labeled as $\mu_{c1}$, $\mu_{c2},$ and $\mu_{c3},$ respectively, which has a magnitude relationship. Therefore, the chemical potential of the trimer must also be set according to the magnitude relationship of the three different chemical potentials in the PMM. In the trimer, the gap *g* between nanodiscs is set as 10 nm, and the chemical potentials of three nanodiscs from left to right are labeled as $\mu_{ca1}$, $\mu_{ca2},$ and $\mu_{ca3}$, respectively. For the structure of N = 19, the chemical potential of the nanodisc has a relationship of $\mu_{c1}\geq \mu_{c3}>\mu_{c2}$ when $\mu_{c3}$ is modified. Therefore, $\mu_{ca2}$ and $\mu_{ca3}$ are set as 0.3 eV and 0.7 eV, respectively, while $\mu_{ca1}$ is 0.3 eV, 0.4 eV, and 0.5 eV, respectively. The near-field distributions of the three different $\mu_{ca1}$ are shown in Figure 3d. With the increase of $\mu_{ca1}$, the electromagnetic field of the left nanodisc is pushed to the middle nanodisc, and the right nanodisc is also weakly enhanced. The behaviour of the leftmost nanodisc is explained by the equivalent refractive index of graphene, and the equation is written as the equation below.
(7)$$n_{eff}=\frac{2i\varepsilon_{r}\varepsilon_{0}c}{\sigma_{g}}$$
where $\varepsilon_{r}=1/2\cdot \left(\varepsilon_{air}+\varepsilon_{CaF_{2}}\right)$ is the average dielectric constant of the environment around graphene nanodiscs. $\varepsilon_{CaF_{2}}$ is the dielectric constant of CaF_2_, which is obtained from the handbook [55]. $\varepsilon_{0}$ and *c* are the vacuum permittivity and the speed of light in vacuum, respectively. The surface conductivity $\sigma_{g}$ is a function of $\mu_{c},$ according to Equation (2). For a given frequency, $n_{eff}$ of graphene decreases as the chemical potential increases [24], which results in the reduction of the confinement ability of the electromagnetic field. However, due to the relationship of $\mu_{ca3}>\mu_{ca1}>\mu_{ca2}$, the confinement ability of the intermediate nanodisk to light is the strongest.

Therefore, increasing the chemical potential of the outer ring nanodiscs mainly affects the interaction between the outer ring and the middle ring. With the increase of $\mu_{c3}$, the electromagnetic field in the outer ring nanodiscs is pushed to the middle ring nanodiscs, which leads to the weakened effect of the outer ring nanodiscs. Hence, the amplitude of peak A decreases and the FR_2_ mode disappears when $\mu_{c3}$ is 0.52 eV. Simultaneously, the amplitude of peak B decreases, which further confirms the conclusion of previous discussions. In Figure 3a,b, it is clear that the extinction spectra have a blue shift with the increase of $\mu_{c3}$. In essence, the adjustment of the chemical potential modifies the plasmon frequency of the graphene nanodisc, which satisfies the equation below [4,37,44].
(8)$$\omega_{pl}=\sqrt{\frac{e^{2}\mu_{c}q}{2\pi \hbar^{2}\varepsilon_{0}\varepsilon_{r}}}\propto \sqrt{\frac{\mu_{c}}{r\varepsilon_{r}}}$$
where the wave vector q=π/2r, and r is the radius of the nanodisc. Therefore, the plasmon frequency increases with the increase of the chemical potential or the decrease of the radius.

The extinction spectra with the variation of radius $R_{3}$ are presented in Figure 3e. As $R_{3}$ decreases, peak A decreases and the FR_2_ mode disappears gradually. When $R_{3}$ is 48 nm, the FR_2_ mode disappears entirely. Simultaneously, the difference between peak B and peak C is similar with the relation between peak a and peak b in the heptamer, which further confirms the analysis in the previous subsection. Due to the large number of the outer ring nanodiscs, some change of $R_{3}$ has a great effect on spectra, which is more clear than the change of the radius in the heptamer and octamer [56].

### 3.3. The Competition between the Two FR Modes via the Intermediate Ring Nanodiscs

For the FRs produced by a noble metal structure, once the structure is determined, it is extremely difficult to modulate FRs except for the modification of the polarization direction of the incident light. Therefore, it is of great significance to flexibly control the FRs produced by a graphene nanostructure with adjustable chemical potential. The chemical potential of all the intermediate ring nanodiscs is adjusted from 0.5 eV to 0.52 eV, and the calculated extinction spectra are shown in Figure 4a. With the increase of $\mu_{c2}$, it is found that the FR_1_ mode degrades gradually, while the FR_2_ mode still exists. When $\mu_{c2}$ varies from 0.5 eV to 0.48 eV, the FR_2_ mode disappears gradually, while the FR_1_ mode still exists, as presented in Figure 4b. Selective excitement and competition between the FRs is feasible by texturing the chemical potential of the intermediate ring nanodiscs, which is significant for the fields of optical devices and biochemical sensors.

In order to explore the influence of chemical potential of the intermediate ring nanodiscs on other nanodiscs, attention is also paid to the simple trimer model, which is schematically shown in Figure 4c, where the chemical potentials of the left and the right nanodisc are fixed to 0.5 eV and 0.6 eV, respectively. The chemical potential of the middle nanodisc is set as 0.5 eV, 0.55 eV, and 0.6 eV, respectively, and the near-field distributions are illustrated in Figure 4d. When $\mu_{ca2}$ is 0.5 eV, the electromagnetic field is mainly distributed on the left nanodisc and the middle nanodisc because the chemical potentials of nanodiscs satisfy a relationship of $\mu_{ca1}=\mu_{ca2}$, while the electromagnetic field distribution of the right nanodisc is weaker than the left and the middle nanodiscs due to its low binding capacity to incident light. With the increase of $\mu_{ca2}$, the chemical potentials meet the relationship of $\mu_{ca3}\geq \mu_{ca2}>\mu_{ca1}$, which makes the electromagnetic field shift to the left nanodisc. The electromagnetic field in the middle nanodisc degrades faster than that in the right nanodisc. Therefore, for the nanostructure of N = 19, when the chemical potential of the intermediate ring nanodiscs is increased, the effect of the intermediate ring nanodiscs is weakened, while the effect of the outer ring nanodiscs is enhanced. This makes the FR_1_ mode disappear rapidly, while the FR_2_ mode is enhanced. However, for the case of the decreasing $\mu_{c2}$, the relationship of chemical potential is $\mu_{c1}>\mu_{c3}>\mu_{c2}$. Based on the analysis above, the electric field distribution in the intermediate ring nanodisks is enhanced with decreasing $\mu_{c2}$, while the field in the outer ring nanodisks is weakened. Thus, the FR_1_ mode is enhanced, and the FR_2_ mode gradually disappears. On the other hand, according to Equation (8), the resonant frequency is proportional to the chemical potential. As the chemical potential increases, the carrier concentration increases and the graphene kinetic inductance decreases, which results in the resonant frequency blue shift. Similarly, a red shift occurs in extinction spectra when $\mu_{c2}$ reduces from 0.5 eV to 0.48 eV.

In this case, the influence of the radius of the intermediate ring nanodiscs on spectra is also considered. A series of extinction spectra with the decrease of $R_{2}$ are displayed in Figure 4e. It demonstrates that the FR_1_ mode disappears gradually when the radius reduces from 50 nm to 47 nm, while the FR_2_ mode always exists in the spectra. This is contrary to the result of the variation of the radius of the outer ring nanodiscs. Decreasing the radius of the intermediate ring nanodiscs not only causes the blue shift of resonant frequency, but also reduces the role of the intermediate ring nanodiscs in the whole structure. Therefore, the FR_1_ mode, which relies on the intermediate ring nanodiscs, disappears completely when the radius decreases to 47 nm.

### 3.4. Modulation Depth of the Tunable FRs

For FR, the degree of detuning between the bright mode and dark mode is illustrated by the contrast between them. In this scenario, we vary the depth by adjusting the parameters of the central nanodisc such as the chemical potential. For the change of chemical potential, the chemical potential of the central nanodisc is adjusted from 0.6 eV to 0.9 eV because only a single nanodisc is considered. Simultaneously, the $\mu_{c2}$ and the $\mu_{c3}$ are kept at 0.5 eV. The extinction spectra with different $\mu_{c1}$ is plotted in Figure 5a. It is interesting that all the peaks are enhanced with the increase of $\mu_{c1}$, which is different from the heptamer and hexamer [24]. For a given wavelength within the range of interest, the refractive index increases with the decreasing of the chemical potential of graphene. Thus, the confinement capability with the identical geometry structure is enhanced with the reduction of the chemical potential. In the process of the variation of $\mu_{c1}$, the relationship between the chemical potential of nanodiscs remains at $\mu_{c1}>\mu_{c2}=\mu_{c3}$. Although the increase of the chemical potential of the central nanodisc weakens the electric field distribution of the central nanodisc, it can also push the electromagnetic hot spots to the surrounding nanodiscs [29], which results in the enhancement of the electric field distribution of the intermediate ring nanodiscs and the outer ring nanodiscs. Furthermore, the electric field distributions of peak A, peak B, and peak C are mainly concentrated on the intermediate ring nanodiscs and the outer ring nanodiscs, which results in the enhancement of all the peaks. Simultaneously, the depth of FRs becomes deeper, which is shown in Figure 5a. The depth of FR is defined as [57] D=P−V, where *P* is the amplitude of the peak and *V* is the amplitude of the dip. The depth of the two kinds of FR modes varies with the chemical potential, as shown in Figure 5b. It is found that the depth of the FR_2_ mode is higher than the FR_1_ mode. When $\mu_{c1}$ is 0.6 eV, the depth of the two FR modes is close to each other. However, with the increase of the chemical potential to 0.9 eV, the depth of the FR_2_ mode reaches 8726.93 nm^2^, and the depth of the FR_1_ mode increases to 5562.59 nm^2^. This phenomenon can be interpreted as the following. With the increase of $\mu_{c1}$, the electric fields of the intermediate ring nanodiscs and the outer ring nanodiscs are enhanced, which also leads to the bright mode and the dark mode of FR get enhanced. However, because the FR_1_ mode is mainly generated by the heptamer, the effect of the outer ring nanodiscs is weaker than the FR_2_ mode. Therefore, when the outer ring nanodiscs are strongly excited, the adjustment of the FR_2_ mode is larger than the FR_1_ mode. Therefore, the depth of the FR_2_ mode is larger than the FR_1_ mode when the bright and dark modes are not detuned. Simultaneously, the quality of FR with the change of $\mu_{c1}$ is also considered, which can be characterized by the quality factor, and the Q factor is described by the following equation [20]: $Q=f_{0}/\delta f$, where $f_{0}$ is the frequency at Fano dip, δf is the full width at half-maximum ( FWHM )of the Fano resonance. However, the bandwidth is defined as the frequency difference between the dip and higher frequency of the peak in the asymmetric Fano resonance. For the FR_2_ mode, the Q factor is 159.67 when $\mu_{c1}$ is 0.6 eV. As $\mu_{c1}$ increases to 0.9 eV, the carrier concentration of graphene nanodisks increases. The broad extinction curves become broader due to the increasing losses, which results in the Q factor decreasing to 95.4, as shown in Figure 5c. Therefore, with the increase of the chemical potential of the central nanodisc, the depth of FR increases, but the quality of FR decreases.

## 4. Conclusions

In summary, a graphene plasmonic metamolecule is constructed by adding 12 nanodiscs to a graphene plasmonic heptamer. Two different modes of Fano resonance are obtained in the extinction spectra of the proposed structure. By the growing of the chemical potential of the outer ring nanodiscs, it is found that the short wave FR_2_ mode is first to be modified, while the FR_1_ mode is slowly adjusted when the chemical potential of the outer ring further increases. Nevertheless, selective excitement and competition between FR_1_ and FR_2_ modes is achieved by the modification of the intermediate ring nanodiscs. A simple trimer model employed to understand the effect of the modification of the chemical potentials in the radial direction of the graphene plasmonic metamolecule. The proposed graphene nanostructures might find broad application in the fields of the high-performance biochemical sensor and nanoantenna.

## Figures and Tables

**Figure 1 nanomaterials-10-00236-f001:**
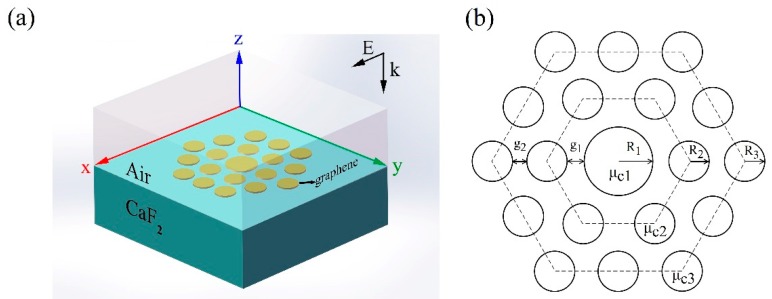
(**a**) Schematic of the graphene metamolecule. Nineteen nanodiscs are placed on a CaF_2_ substrate and are surrounded by the air. The incident light is polarized along the x-axis. (**b**) The top view of the graphene metamolecule: gap $g_{1}$ = 10 nm, gap $g_{2}$ = 30 nm, radius $R_{1}$ = 90 nm of the central nanodisc, radius $R_{2}$ = 50 nm of the intermediate ring nanodiscs, radius $R_{3}$ = 50 nm of the outer ring nanodisks, chemical potential $\mu_{c1}$ = 0.6 eV, chemical potential $\mu_{c2}$ = 0.5 eV, and chemical potential $\mu_{c3}$ = 0.5 eV.

**Figure 2 nanomaterials-10-00236-f002:**
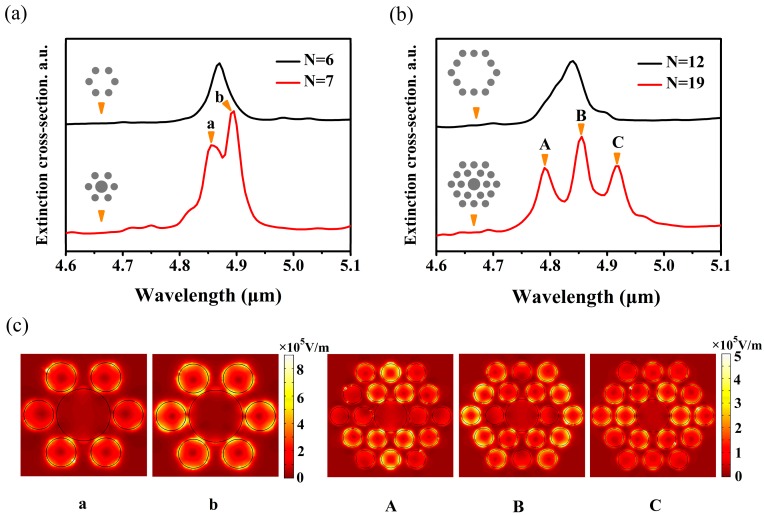
(**a**) Extinction spectra of hexamer (black) and heptamer (red). (**b**) Extinction spectra of the outer ring nanostructure (black) and PMM structure with 19 nanodiscs (red). (**c**) The normalized electric field distributions (|E|) of peak a, b, A, B, and C.

**Figure 3 nanomaterials-10-00236-f003:**
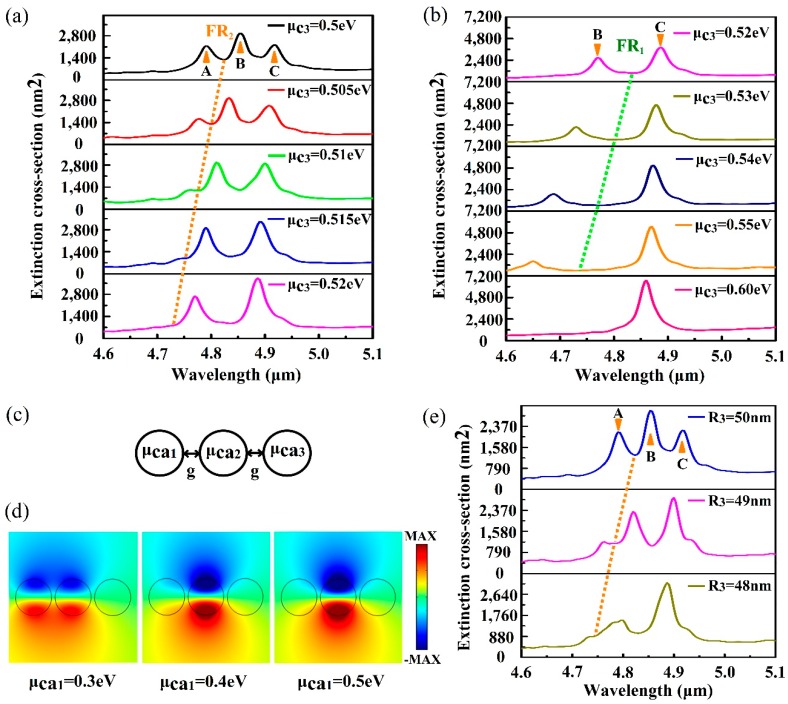
(**a**) Extinction spectra with the increase of $\mu_{c3}$ ranged from 0.5 eV to 0.52 eV. (**b**) Extinction spectra with the increase of $\mu_{c3}$ ranged from 0.52 eV to 0.6 eV. (**c**) Schematic of the graphene trimer nanostructure. (**d**) The near-field distributions with the variation of $\mu_{ca1}$. (**e**) Extinction spectra with the change of radius $R_{3}$.

**Figure 4 nanomaterials-10-00236-f004:**
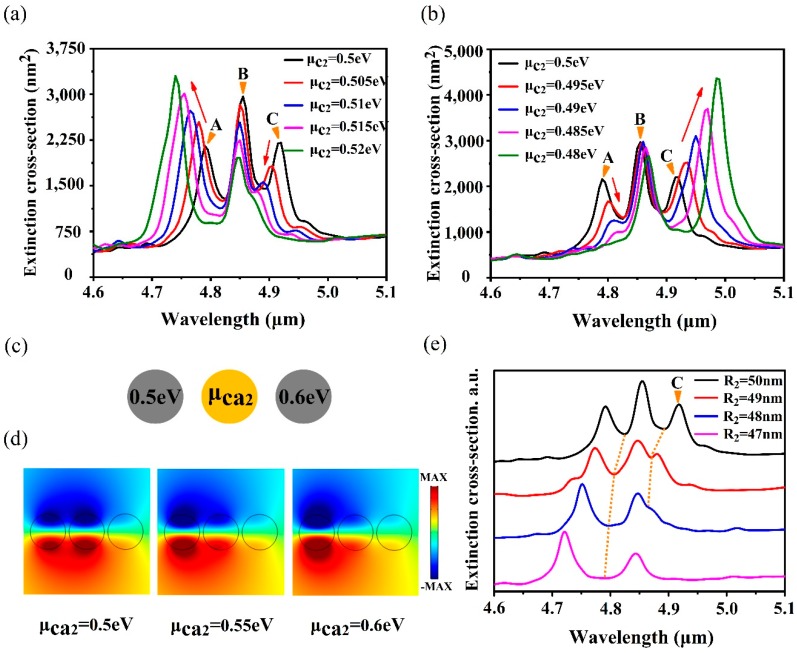
(**a**) Extinction spectra with the adjustment of $\mu_{c2}$ from 0.5 eV to 0.52 eV. (**b**) Extinction spectra with the variation of $\mu_{c2}$ from 0.48 eV to 0.5 eV. (**c**) The distribution of the chemical potential in the trimer. (**d**) The near-field distributions with different $\mu_{ca2}$. (**e**) The influence of the radius of the intermediate ring nanodiscs on FR modes.

**Figure 5 nanomaterials-10-00236-f005:**
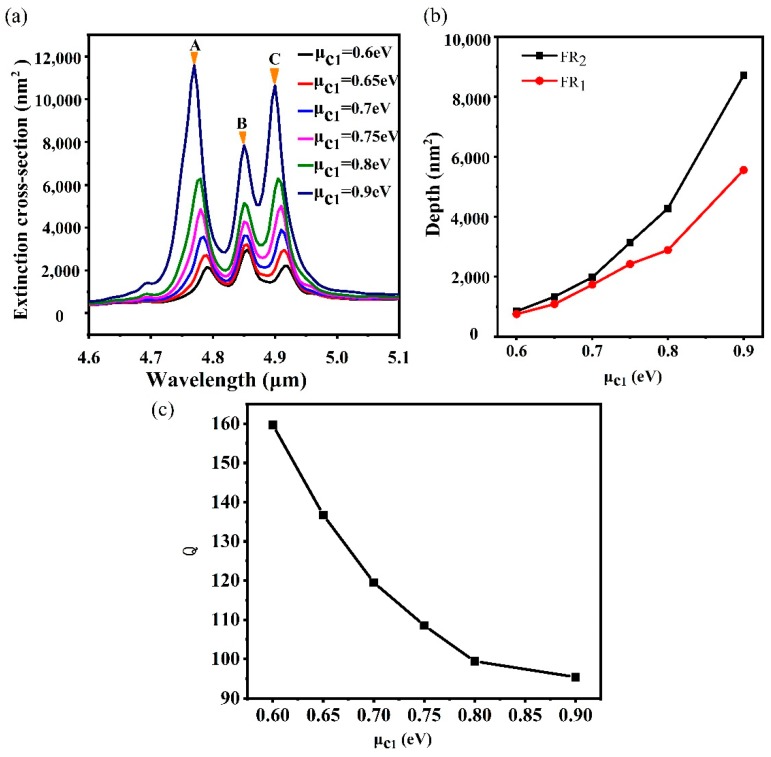
(**a**) Extinction spectra with the increase of the chemical potential of the central nanodisc. (**b**) Modulation depth of two FR modes with the variation of $\mu_{c1}$. (**c**) The Q factor of the FR_2_ mode with the variation of $\mu_{c1}$.

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
