# Peer review of "Multiple Fano Resonances with Tunable Electromagnetic Properties in Graphene Plasmonic Metamolecules"

_nanomaterials, 2020, doi:10.3390/nano10020236_

Round 1

Reviewer 1 Report

An interesting theoretical paper that could gain greater impact if various English language errors were corrected and some reference to experimental work were to be included.

Errors include:

Page 3: Matemolecule?

"Further, a simple trimer model is employed to emulate the radical direction of the whole structure"

"The proposed PM consisting of nineteen graphene nanodisks with a D6h point group symmetry 80 is placed on a calcium fluoride"

"In the infrared region, SiO2 substrate produces strong phonon-related resonance peaks,"

Etc etc

Author Response

Dear Editors and Reviewers: 

Thank you for your comments concerning our manuscript entitled “Multiple Fano Resonances with Tunable Electromagnetic Properties in Graphene Plasmonic Metamolecules”. These comments are all valuable and very helpful for revising and improving our paper, as well as the important guiding significance to our research. We have made revision according to the reviewers’ comments. The main corrections in the paper and the responds to the reviewer’s comments are as flowing:

Comment 1: An interesting theoretical paper that could gain greater impact if various English language errors were corrected and some reference to experimental work were to be included.

Response: Thanks for your serious review. We have corrected various English language errors in the paper. As for the reference to experimental work, several articles have been cited, such as Ref. (1), Ref. (5), Ref. (6), Ref. (48), Ref. (49), Ref. (51), Ref. (54) and Ref. (56).

Comment 2: Page 3: Matemolecule?

Response: Thanks for your serious review. We rewrote this word.

Comment 3: "Further, a simple trimer model is employed to emulate the radical direction of the whole structure”

Response: Thanks for your serious review. As the reviewer’s comment, we add this word in the revised manuscript.

Comment 4: "The proposed PM consisting of nineteen graphene nanodisks with a D6h point group symmetry 80 is placed on a calcium fluoride"

Response: Thanks for your serious review. According to the reviewer’s comment, we rewrote this sentence in the revised version.

Comment 5: "In the infrared region, SiO2 substrate produces strong phonon-related resonance peaks,”

Response: Thanks for your serious review. As the reviewer’s comment, we rewrote this sentence in the revised version.

Special thanks to you for your good comments.

Reviewer 2 Report

This paper is vey well written and it is suitable for publication on this international journal but only after inor revisions, listed below:

a) Please, Could the authors add a real case of study, showing the corresponding graphene samples?

b) After to show point a, Please, Could the authors add the characterization study of this graphene sample? As: Raman, ATR, XPS, HR-TEM and AFM imaging

Author Response

Dear Editors and Reviewers: 

Thank you for your comments concerning our manuscript entitled “Multiple Fano Resonances with Tunable Electromagnetic Properties in Graphene Plasmonic Metamolecules”. These comments are all valuable and very helpful for revising and improving our paper, as well as the important guiding significance to our research. We have made revision according to the reviewers’ comments. The main corrections in the paper and the responds to the reviewer’s comments are as flowing:

Comment 1: This paper is vey well written and it is suitable for publication on this international journal but only after inor revisions, listed below:

a) Please, Could the authors add a real case of study, showing the corresponding graphene samples? b) After to show point a, Please, Could the authors add the characterization study of this graphene sample? As: Raman, ATR, XPS, HR-TEM and AFM imaging

Response: Thanks for your serious review. At the same time, thank you very much for your affirmation of our manuscript. In our current work, we only study the characteristics of the graphene Metamolecule numerically, but we haven’t done experimental analysis. In the future work, we will further study the plasmonic properties of the graphene metamolecules by experiment.

Special thanks to you for your good comments.

Reviewer 3 Report

In the given manuscript, Zhou et al. construct graphene plasmonic metamolecule (PMM) which is composed of graphene heptamer and surrounded 12 graphene nanodisks to support two Fano resonance modes in extinction spectra without spatial symmetry change. The authors performed simulation for the graphene PMM on CaF2 substrate and adjusted the chemical potential of outer ring, intermediate ring, and central nanodisks, respectively, to observe the effect on the Fano resonance. Unfortunately, the submitted manuscript is insufficient for publication in Nanomaterials because the overall experimental design seems to be not reasonable: graphene PMM with D6h point group symmetry is considered to support Fano resonance, why is particularly trimer model introduced to analyze the extinction spectra and electric field distribution for varied chemical potential of the graphene? The author should present an evidence for the link bewteen them or support by the data only for graphene PMM. Followings are minor points I’d like to suggest, to improve the manuscript.

Author Response

Dear Editors and Reviewers: 

Thank you for your comments concerning our manuscript entitled “Multiple Fano Resonances with Tunable Electromagnetic Properties in Graphene Plasmonic Metamolecules”. These comments are all valuable and very helpful for revising and improving our paper, as well as the important guiding significance to our research. We have made revision according to the reviewers’ comments. The main corrections in the paper and the responds to the reviewer’s comments are as flowing:

Comment 1: In Fig. 1a, ‘graphene’ would be better to be labeled like air and CaF2. In Figure caption, meaning of the μc in Fig. 1(b) would be better to be mentioned like R or g.

Response: Thanks for your serious review. As your comment, the ‘graphene’ has added into Fig. 1a to be labeled. Furthermore, the meaning of the μc has been mentioned in Figure caption.

Comment 2: What is the value of μc3 in Fig. 2b? What is the value of μc2 and μc3 in Fig. 5?

Response: Thanks for your serious review. The value of μc3 in Fig. 2b has been mentioned. Furthermore, the value of μc2 and μc3 in Fig. 5 have given in our revised manuscript.

Comment 3: The authors describes that lineshape of the FR mode consists of two peaks and a dip. If so, does FR1 mode indicate the region from peak C to peak B (including one dip), and FR2 mode indicate the region from peak B to peak A (including one dip)? Since these modes don’t indicate the peaks labeled in Figures, readers can be confused.

In Fig. 3a, it would be better to label yellow dotted line as FR2 mode if correct. Description of blue shift of peak B in Fig. 3b does not exist (instead FR1 mode is mentioned), so I think the yellow dotted line in Fig. 3b would be better to removed. Instead, I’d like to suggest new dotted indicating FR1 mode with different color from FR2 mode of Fig. 3a. The dotted line in Fig. 3f would be better to indicate FR2 mode, similar to Fig. 3a

Response: We have added more information in the revised paper. We mark two modes respectively in Fig. 3a and Fig. 3b, and modify Fig. 3e according to your opinions.

Comment 4: Fig. 3c and Fig. 3d could be merged with each other.

Response: We have merged F ig. 3c and Fig. 3d with each other.

Comment 5: In Fig.3a and Fig. 3b, μc1=0.6 eV, μc2=0.5 eV, and μc3=0.5~0.6 eV. But in line 204 on p6, μca1=0.3, 0.4, and 0.5 eV, μca2=0.3 eV, μca3=0.7 eV. Why do the chemical potentials of the nanodisks in the trimer have large difference compared to original graphene PMM? Is it only to see the significant change in electric field distribution? If so, the analysis might be incorrect.

Response: Thanks for your serious review. We use the trimer to explore the effect of changes in chemical potential on nanodisks, which has indicated in paper. In our previous studies, we found that the effect on the nanodisk can be ignored when the chemical potential of a single nanodisk is changed slightly, but the impact on the surrounding nanodisks needs to be further explored. Here, the trimer is used to explore the influence of different position of nanodisk on other nanodisks with the change of chemical potential, so as to get the regular of its change. Because the change regular of chemical potential in a single nanodisk is difficult to obtain in a small range, the gap between chemical potential changes is increased. Furthermore, the effect of changing the chemical potential of the trimer in the range of 0.3eV to 0.7eV is better than that in the range of 0.5eV to 0.6eV. Moreover, compared with the trimer, the slight change of the chemical potential of the outer ring nanodisks lets to a great impact because of the number of the outer ring nanodisks. Therefore, in order to get a more obvious regular of variation, there is a big difference between the chemical potential of trimer and that of PMM.

Comment 6: Is only analysis that “a little change of R3 has a great effect on spectra” in line 231 on p7 for Fig. 2f? In my opinion, the change in the extinction spectra due to the reduced R3 can be compared to the change due to the increase of μc3.

Response: As the reviewer’s comment, the change in the extinction spectra due to the reduced R3 can be compared to the change due to the increase of μc3 because of the large number of the outer ring nanodisks. However, a little change of R3 has a great effect on spectra, which is compared with the results of other scholars. Therefore, we have added more information into this sentence in the revised version.

Comment 7: In line 255 on p8, what does “right nanodisk is weak”?

Response: Thanks for your serious review. In the revised version, we have added more information into this sentence to make it more clear.

Comment 8: In line 270 on p9, the authors mention that “The radius of the intermediate ring nanodisks is reduced to avoid the electron tunneling when the radius of nanodisk increases”. What does that mean? Can’t same effect exist in Fig. 3f? In my opinion, similar to Fig. 3f, the effects of intermediate nanodisks on spectra might decrease as their size decreases, even the behavior of FR modes in Fig. 4e is contrary to that in Fig. 3f.

Response: The electron tunneling usually occurs when the gap between particles is very small, which has been studied in the paper “Bridging quantum and classical plasmonics with a quantum-corrected model”. The quantum-mechanical effects due to interparticle electron tunneling can weaken the coupling strength between particles. Therefore, in order to avoid this effect, we reduce the radius to increase the distance between particles. Here, the radius of the intermediate ring nanodisks and the outer ring nanodisks are reduced in order to avoid the phenomenon of electron tunneling. However, the influence of the radius changes of the intermediate ring nanodisks and the outer ring nanodisks on the two kinds of FR is opposite, which is caused by their different contributions to the two FR modes. Therefore, the behavior of FR modes in Fig. 4e is contrary to that in Fig. 3f. Furthermore, it can be obtained that the effects of intermediate nanodisks and the outer ring nanodisks on spectra decrease gradually as their size decreases from Fig. 3f and Fig. 4e, which is due to the fact that the interaction degree of plasmons is not linear with the distance between particles.

Comment 9: In line 288 on p9, since μc1>μc2=μc3, the electric field of intermediate and outer ring nanodisks are enhanced and consequently all the peaks are enhanced. In other words, only refractive index of central nanodisk is decreased. In this situation, overall effective refractive index is also decreased and the confinement effect is typically degraded. Therefore, the authors need to explain it more details that, nevertheless, why all the peaks are more enhanced.

Response: Thanks for your serious review. We have explained that why all the peaks are more enhanced with the increase of μc1 in paper.

Comment 10: In the lower part on p9, the authors discuss Q factor of the spectra in Fig. 5a. I’d like to recommend mentioning Q factor for FR1 mode and visualizing the Q factor of two FR modes with the variation of μc1, similar to the modulation depth in Fig. 5b

Response: As the chemical potential increases, there is a blue shift in both FR modes, as shown in Fig. 5a. Moreover, due to the close resonance frequency of the two FR modes, there is crosstalk between the two modes, which is illustrated in the paper “Higher order Fano graphene metamaterials for nanoscale optical sensing, Nanoscale, 2017 Oct 12;9(39):14998-15004”. In this case, the Q factor of the FR1 mode is affected greatly by the chemical potential. Therefore, we do not discuss the Q factor of the FR1 mode. However, the higher frequency of peak of FR2 mode does not affect by FR1 mode. Simultaneously, we have visualized the Q factor of FR2 mode with the variation of μc1.

Comment 11: Minor corrections.

Between some words there are double spaces. Line 110 on p3: ‘and’ needs to be removed. Line 113 on p3: repeated phrase of ‘the closed surface of’ needs to be removed. Line 285 on p9: phrase ‘decreasing of the graphene’ needs to be corrected

Response: Thanks for your serious review. We have made changes according to your suggestions.

Thank you very much for your valuable advice.

Round 2

Reviewer 3 Report

I’d like to appreciate authors’ great effort in the submitted revision. I think almost minor corrections were performed well, but the important questions still were not solved.

Author Response

Dear Editors and Reviewers: 

Thank you for your letter and for the reviewers’ comments concerning our manuscript. We have made revision according to the reviewer’s comments. The main corrections in the paper and the responds to the reviewer’s comments are as flowing:

Referee #3:

Comment 1: In the previous comments, I’ve mentioned that “why is particularly trimer model introduced to analyze the extinction spectra and electric field distribution for varied chemical potential of the graphene? The author should present an evidence for the link between them or support by the data only for graphene PMM.” prior to all the numbered minor comments. I’m suspicious of equaivalence between the simulation results of trimer and that of whole 19 nanodisk graphene structure. As seen in panel A, B, and C of Fig. 2c, the electric field is not uniform in tangential direction (unlike panal a and b of Fig. 2c). In other words, the electric field distribution of top two nanodisks at intermediate ring (hot spot position) is different from that of middle one at intermediate ring. Also, the field distribution of top center nanodisk at the outer ring is different from that of top left and right nanodisks at the outer ring. Therefore, I don’t think that left panel of Fig. 3e (μca1=0.3 eV,  μca2=0.3 eV,  μca3=0.7 eV) can support panel A, B, and C of Fig. 2c (μca1=0.6 eV,  μca2=0.5 eV, μca3=0.5 eV) without consideration of the absolute value of the chemical potential. Why do not the authors show the electric field of 19 nanodisks with increase of the chemical potential?

Response: Thanks for your serious review. In this paper, PMM is divided into three parts: the outer ring nanodisks, the intermediate ring nanodisks and the central nanodisk. Moreover, the influence of change in the chemical potential of one part of the nanodisks on the other parts is considered. Therefore, in order to demonstrate the effect of modification in the chemical potential of the surrounding nanodisks, we introduce the trimer model to emulate the structure of the graphene metamolecule in the radical direction. The chemical potential distribution from left to right in the trimer corresponds to that of the center, middle and outer ring of the metamolecule. Here we only concern the effect of the changing chemical potentials on the electromagnetic (EM) field distribution in the whole structure, which eventually determines the Fano resonance spectra. Moreover, we believe the mechanism of the modification of the EM field distributions is identical in both the trimer and the radical direction of the PMM. Therefore, we set the chemical potential of the trimer according to the relation in the PMM. Moreover, the idea of using the simple oligomers to explain complex oligomers has been proved to be feasible (see paper “Ren J, Wang W, Qiu W, et al. Dynamic tailoring of electromagnetic behaviors of graphene plasmonic oligomers by local chemical potential[J]. Physical Chemistry Chemical Physics, 2018, 20(24): 16695-16703.”).

In the comments, the reason of uneven electric field distribution of the nanodisks at ring is, that the addition of the outer ring nanodisks results in the coupling and hybridization of plasmonic modes, which further makes the electric field distribution of the nanodisks at ring uneven (see paper “Dregely D, Hentschel M, Giessen H. Excitation and tuning of higher-order Fano resonances in plasmonic oligomer clusters[J]. ACS nano, 2011, 5(10): 8202-8211.”). Although the electric field distributions of the nanodisks are uneven, the changes of chemical potential of the nanodisks at ring have the same effect on the nanodisks at other rings (see paper “Ren J, Wang G, Qiu W, et al. A flexible control on electromagnetic behaviors of graphene oligomer by tuning chemical potential[J]. Nanoscale research letters, 2018, 13(1): 349.”). Thus, the principle of the change of chemical potential in the trimer still supports the phenomenon of the PMM. Owing to the coupling of plasmons in the PMM, the electric field strength of some nanodisks is greatly enhanced, while the other nanodisks are weakly enhanced, which leads to the change of some nanodisks is not obvious in the whole electric field distribution when the chemical potential changes. Therefore, we do not show the electric field distribution of the PMM with the change of chemical potential, but introduce the trimer model to explain the principle produced by the change of chemical potential.

Comment 2: As the authors responded to previous comment 5, range of 0.3 eV to 0.7 eV can give undoubtedly obvious effect than the range of 0.5 to 0.6 eV. But I don’t think the chemical potential difference by 0.3 eV for trimer and that by 0.1 eV for 19 nanodisks are equivalent. If the authors consider the chemical potential difference, instead of the absolute value, why don’t the authors choose μca1=0.5 eV, μca2=0.5 eV, μca3=0.9 eV?

Response: Thanks for your serious review. As mentioned in your comment, the chemical potential difference by 0.3 eV for trimer and that by 0.1 eV for 19 nanodisks are hard to compare. However, the effect of changes in the chemical potential of the nanodiscs on the surrounding nanodisks does not change accordingly, and such changes have been studied (see paper “Ren J, Wang W, Qiu W, et al. Dynamic tailoring of electromagnetic behaviors of graphene plasmonic oligomers by local chemical potential[J]. Physical Chemistry Chemical Physics, 2018, 20(24): 16695-16703.”). Moreover, the introduction of the trimer is also to study the influence of the change of chemical potential on other nanodisks at different positions, and to find out the mechanism of competition of the multiple Fano resonance. At the same time, it also makes the magnitude relationship of chemical potential of the trimer meet that of the three parts in PMM, i.e. the chemical potential of the left nanodisk is between that of the middle nanodisk and that of the right nanodisk and the chemical potential of the outer ring nanodisks is between that of the intermediate ring nanodisks and that of the central nanodisk. Here, the chemical potential of the left nanodisk of the trimer must be between that of the middle nanodisk and that of the ring nanodisk. Therefore, μca1=0.5 eV, μca2=0.5 eV, μca3=0.9 eV is also suitable for the trimer, but μca1 must be between 0.5eV and 0.9eV when μca1 increases.

Comment 3: Related to comment 8 in previous comments, the meaning of the sentence “The radius of the intermediate ring nanodisks is reduced to avoid the electron tunneling when the radius of nanodisk increases” is not clear: radius is reduced and increases again. The authors mentioned that they reduced the radius to increase the distance between particles. As the authors said, it is obvious that the electron tunneling can be prevented when the distance between the particles are increased. If so, I think the origin of the effect could be complicated: the effect from long distance or from small radius itself. Also, the prevention effect of the electron tunneling would be included in Fig. 3f. I wonder what if the radius of nanodisks decreases without change of gap distance.

Response: We delete this sentence to avoid unclear expression.

Thank you very much for your advice.